# Idiosyncratic Drug-Induced Liver Injury (DILI) and Herb-Induced Liver Injury (HILI): Diagnostic Algorithm Based on the Quantitative Roussel Uclaf Causality Assessment Method (RUCAM)

**DOI:** 10.3390/diagnostics11030458

**Published:** 2021-03-06

**Authors:** Rolf Teschke, Gaby Danan

**Affiliations:** 1Department of Internal Medicine II, Division of Gastroenterology and Hepatology, Klinikum Hanau, Academic Teaching Hospital of the Medical Faculty, Goethe University Frankfurt/ Main, D-63450 Hanau, Germany; 2Pharmacovigilance Consultancy, F-75020 Paris, France; gaby.danan@gmail.com

**Keywords:** artificial intelligence (AI), CAM, diagnostic algorithm, idiosyncratic drug-induced liver injury (DILI), herb-induced liver injury (HILI), quantitative RUCAM, Roussel Uclaf Causality Assessment Method (RUCAM)

## Abstract

Causality assessment in liver injury induced by drugs and herbs remains a debated issue, requiring innovation and thorough understanding based on detailed information. Artificial intelligence (AI) principles recommend the use of algorithms for solving complex processes and are included in the diagnostic algorithm of Roussel Uclaf Causality Assessment Method (RUCAM) to help assess causality in suspected cases of idiosyncratic drug-induced liver injury (DILI) and herb-induced liver injury (HILI). From 1993 until the middle of 2020, a total of 95,865 DILI and HILI cases were assessed by RUCAM, outperforming by case numbers any other causality assessment method. The success of RUCAM can be traced back to its quantitative features with specific data elements that are individually scored leading to a final causality grading. RUCAM is objective, user friendly, transparent, and liver injury specific, with an updated version that should be used in future DILI and HILI cases. Support of RUCAM was also provided by scientists from China, not affiliated to any network, in the results of a scientometric evaluation of the global knowledge base of DILI. They highlighted the original RUCAM of 1993 and their authors as a publication quoted the greatest number of times and ranked first in the category of the top 10 references related to DILI. In conclusion, for stakeholders involved in DILI and HILI, RUCAM seems to be an effective diagnostic algorithm in line with AI principles.

## 1. Introduction

Idiosyncratic drug-induced liver injury (DILI) and herb-induced liver injury (HILI) require innovation, although they continuously attract international interest [1,2,3,4,5,6,7,8,9,10,11,12], with a focus on various topics such as mechanistic steps in DILI [1] including unresolved issues [2], the scientometric evaluation of the global knowledge base of DILI highlighting Roussel Uclaf Causality Assessment Method (RUCAM) [3] and its authors [4], and diagnostic aspects of DILI and HILI cases using RUCAM [5,6,7]. RUCAM was also viewed as a valuable diagnostic algorithm based on the principles of artificial intelligence (AI) [8] and appreciated by providing additional cases of DILI assessed for causality by RUCAM [9,10]. A new list of top drugs implicated in DILI assessed by RUCAM clarified issues [11] raised by the disputable list provided by the LiverTox database, which did not use RUCAM to assess causality for its DILI cases but a diagnostic approach based on a subjective assessment [12,13]. In detail, drugs causing suspected DILI received a high causality grading if they were included in a large number of published case reports, ignoring case data quality and a correct exclusion of possible alternative causes [10,11,12,13]. On the other hand, good news came from the section of controversies, because a long-standing discussion on HILI caused by green tea extracts was settled by using RUCAM [14,15].

Several other issues relating to both DILI and HILI are ongoing and new promotional ones have emerged. For instance, the importance of a good causality assessment has also been recommended for other database studies on DILI [16], which shows that current issues are not confined to the LiverTox database [10,11,12,13]. It is critical that the common practice within the scientific DILI and HILI community is to use poorly documented cases of the LiverTox database to attempt feature descriptions such as of the risk factors of drugs or herbs causing liver injury. Under these critical conditions, LiverTox requires substantial revision to become an appraised tool of liver injury case details. In addition, problematic diagnostic biomarkers of DILI, which were initially provided and promoted by the European Medicines Agency (EMA) and the US FDA (Food and Drug Administration), lost regulatory support as discussed in 2019 [17] and confirmed in 2020, with additional details regarding retracted publications of a group from Europe [10]. The previous Letters of Support of EMA and the FDA have officially and correctly been retracted due to misconducted external studies [17]. As a consequence, biomarkers lost their credibility due to institutional regulatory shortcomings including misconceptions, but biomarkers are still used and discussed in various publications, ignoring the retraction issue [10].

In this review article, we analyzed idiosyncratic DILI and HILI cases with respect to a causality assessment based on the quantitative RUCAM as the worldwide most used diagnostic algorithm, established early in line with the principles of AI. Related to RUCAM, other new aspects focused on the scientometric evaluation of the global knowledge base of DILI, its utility in the liver injury of patients with COVID-19 infections treated with drugs or herbs, and its mandatory use in DILI and HILI cases to establish new risk factors or mechanistic steps. Finally, and in line with RUCAM, AI-based algorithms are being applied in many other complex diseases for diagnostic and therapeutic reasons.

## 2. Literature Search and Source

The PubMed database was searched for articles by using the following key terms: AI; algorithms; artificial intelligence; idiosyncratic drug-induced liver injury (DILI); herb-induced liver injury (HILI); Roussel Uclaf Causality Assessment Method (RUCAM). Limited to the English language, publications from each search terms were analyzed for suitability of this review article. Publications were complemented by the large private archive of the authors. The final compilation consisted of original papers, consensus reports, and review articles with the most relevant publications included in the reference list of this review.

## 3. Definitions

### 3.1. RUCAM-Based Liver Injury

The liver injury suspected to be caused by drugs and herbs is defined by threshold values of the liver tests (LTs) alanine aminotransferase (ALT) and/or alkaline phosphatase (ALP). Serum activities considered as relevant are for ALT ≥ 5 × ULN (upper limit of normal) and ALP ≥ 2 × ULN, provided that ALP is of hepatic origin [18]. The ALT values eliminate cases without clinical relevance. The original RUCAM was the first causality assessment method (CAM) ever considering threshold criteria, although the values initially proposed for ALT were lower [4] as compared to the currently used criteria [18]. Of note, patients with suspected DILI or HILI, who show LT values below the thresholds, are considered to have no clinically significant liver injury but liver adaptation or tolerance [19]. Regrettably, published case reports on DILI or HILI occasionally contain cases without mentioning ALT or ALP values, excluding these as liver injury cases. Although RUCAM-based threshold values are commonly used in publications on DILI and HILI, credit was not always given to the authors. The current threshold values [18] seem to be generally well accepted worldwide [6,7,8,19,20] and correctly performed using the updated RUCAM [20]. In addition to the RUCAM-related threshold basics it was only a little step considering bilirubinemia to predict a more serious outcome of liver injury. However, bilirubin is not part of the diagnostic RUCAM algorithm and will not be considered further in the current analysis. Likewise, liver histology is not included as an element in RUCAM [4,18], because there are no features typical of DILI and HILI. If liver biopsy was performed in patients with suspected DILI or HILI, the indication was rarely outlined in the publication, and the histology report often concluded that the findings were compatible with an injury induced by a drug, herb or another product, without any diagnostic value. In essence and as a reminder, defining RUCAM-based liver injury is essential, because this has occasionally been forgotten in published reports invalidating the conclusions presented. Details of liver adaptation and liver injury are listed (Table 1).

### 3.2. RUCAM-Based Liver Injury Pattern

RUCAM was also the first CAM differentiating patterns of liver injury, based on LTs without the need for liver histology [4]. The LT criteria of liver injury pattern remained unchanged and are part of the updated RUCAM as described in detail [18]. To determine the types of liver injury, it is easy to calculate the R value, which stands for the ratio of ALT divided by ALP [4,18], best done at the beginning of the liver injury. In detail, the serum activities of ALT and ALP are expressed as a multiple of ULN, respectively, provided that ALP is of hepatic origin. To be used for causality assessment by the updated RUCAM, two types of liver injury are defined: first, a hepatocellular injury with R > 5, and second, a cholestatic/mixed liver injury with R ≤ 5. As the criteria are different in the two types of injury and variable, two scales of RUCAM are available, one for the hepatocellular injury and another one for the cholestatic/mixed liver injury [18]. Thus, determining the liver injury pattern is necessary to proceed with the correct RUCAM scale.

### 3.3. DILI

By convention, DILI is caused by chemical drugs leading to idiosyncratic or intrinsic liver injury [7,19]. Idiosyncratic liver injury is due to the interaction between the drug used in recommended daily doses and a susceptible individual, whereas intrinsic liver injury is caused by drug overdose like acetaminophen (Table 1) [7]. Interesting but not commonly accepted is a third category, provisionally called indirect liver injury, which would result from the medication’s actions rather than from its toxicity, certainly disputable due to lack of a clear disease definition [21]. This category does not prevent using the updated RUCAM [18]. Therefore, in the present analysis, only the idiosyncratic injury form was considered for DILI.

### 3.4. HILI

HILI is due to the use of a variety of products such as regulatory approved herbal drugs, nonregulated herbal medicines [5,6,7], and so called herbal dietary supplements, an incorrect term that should be avoided as they do not supplement any diet [22]. Classifying HILI as idiosyncratic or intrinsic liver injury is hampered by the fact that recommended daily doses for herbal products are not commonly available [5,6,7]. Considering adulterants and contaminants also found in herbal products, in addition to toxic heavy metals due to soil pollutions, mycotoxins, and herbicides as well as pesticide residues occurring during collection, processing and production [23], it is often not possible to ascribe the liver injury to a specific ingredient. However, it remains to be established on a case-by-case basis whether these potential toxins are really toxic to the liver, because RUCAM assesses the causality of the whole herbal product and not of single ingredients. Another problem is the tendency, especially in Asian countries, to classify HILI as DILI, leading to the inclusion of HILI cases in a cohort of DILI cases [5,6,7]. As a result, this mix should be discouraged as it will not allow for a feature description of DILI separately from HILI. The uniformity of study cohorts provides the best results and prevents useless discussion on vague conclusions.

## 4. Historical Background of RUCAM and Call to Name RUCAM Correctly

In the late eighties and due to the fast development of pharmacovigilance activities, it was increasingly recognized that problems exist with the correct diagnosis of suspected DILI cases. Therefore, in order to define terms used in DILI assessment and qualitative criteria based on the French CAM, consensus meetings of experts have been organized by the French Roussel Uclaf pharmaceutical company since 1985, as summarized recently [18]. The first results of the meetings were published in 1988 as criteria to be used with the French CAM [24] and in 1990 as a modified version, which included additional criteria and ascribed qualitative weight to each criterion [25]. Consensus was achieved among the eight experts in hepatology from six countries: J.P. Benhamou (France), J. Bircher (Germany), G. Danan (France), W.C. Maddrey (USA), J. Neuberger (UK), F. Orlandi (Italy), N. Tygstrup (Denmark), and H.J. Zimmerman (USA) [25]. These experts evaluated DILI cases according to characteristics including chronological criteria, risk factors such as age and alcohol use, re-exposure criteria, hepatotoxicity criteria, and liver injury pattern; they standardized DILI case assessment with specific items and received appropriate credit for their contribution [25]. RUCAM was published later after fine-tuning [4,26], and the quantitative weight of each criterion and two-fold validation process by an independent team of Roussel Uclaf for reproducibility and cases including positive rechallenge as a gold standard for performance indicators [26]. Consequently, this approach ensured adequate method validation.

Although clearly named RUCAM for the Roussel Uclaf Causality Assessment Method [4], some authors erroneously changed the name to CIOMS/RUCAM or CIOMS, possibly going back to the consensus meeting in 1989, which was held in Paris under the auspices of Council for International Organizations of Medical Sciences (CIOMS), but the meeting was actually organized by the French pharmaceutical Roussel Uclaf company. Although some qualitative criteria were agreed upon in this meeting, the algorithm, the quantitative score, and the validation process of the method were prepared and executed by the Roussel Uclaf team. Thus, the correct name is and remains RUCAM [4,18], which is now available as updated version, to be used for future cases of suspected DILI and HILI [18]. By definition, RUCAM represents an algorithmic structured diagnostic method, specific to liver injury, based on defined criteria of LT thresholds and liver injury pattern, operating quantitatively using key elements with individual scores, and provides objective final causality gradings [18]. In sum, RUCAM is the correct term.

## 5. Diagnostic RUCAM Algorithm and Artificial Intelligence

AI is closely connected with RUCAM, discussed as a certainly provocative issue in a recent editorial [8]. Mainstream opinion considers AI as en vogue in many areas of human society and the healthcare community [27] including medicine [28,29,30,31,32,33,34]. In disease diagnosis, the inclusion of AI may be beneficial by improving efficacy, accuracy, and precision, and by decreasing workload, not to forget providing a better case for monitoring and saving the budget [34]. At the political level, the European Commission summarized the current state in a White Paper on AI issues released on 19 February 2020, discussing various AI concepts that revolutionized many complex processes [27]. Initial tools were algorithms, and more recently software programs are also being used with increasing frequency [27,35,36]. AI as a special term was created in 1956, when John McCarthy, a professor of Mathematics at Darmouth College, initiated a research project [35] with the aim to reduce complexity by simplifying complex processes found in many areas of the social and technical community. The principle was to provide tools enabling the input of well-defined data into a black box that systematically evaluates and coordinates incoming data, and fosters the output of clear results such as diagnoses in complex diseases [36]. At the time when AI concepts had been developed, the focus was on mostly manually applied algorithms, prior to helpful software availability.

Innovations in medicine supported by AI can help diagnose complex diseases for the sake of patients, physicians, society, and the economy. In a clinical setting, for instance, problems may emerge in patients with a challenging diagnosis that could be overcome with diagnostic algorithms now available for a broad range of diseases, conditions which can help establish the correct diagnoses and timely suggest therapy modalities [4,26,37,38,39,40,41,42,43,44,45,46,47,48,49,50,51,52,53,54,55,56,57,58,59,60,61,62,63]. Most of the diagnostic algorithms were published in Europe, which appears more interested in clarifying clinical issues, whereas US scientists seemed for unknown reasons to be more cautious with establishing innovative approaches like such diagnostic or therapeutic algorithms (Table 2). Under some conditions depending on the concerned disease, diagnostic algorithms represent validated tools because they are composed of key elements that are well defined and individually scored. They also provide a final result based on the sum of individual scores. Establishing an algorithm and applying it under field conditions and the eyes of peers may occasionally be cumbersome and frustrating. Despite these restrictions, performance indicators are good for the majority of the listed algorithms and highly appreciated by the scientific community, with little criticism by peers (Table 2) [4,26,37,38,39,40,41,42,43,44,45,46,47,48,49,50,51,52,53,54,55,56,57,58,59,60,61,62,63].

Considering the diagnostic algorithms using principles of AI, the RUCAM algorithm was among the first ones established worldwide and applicable to DILI (Table 2), published with the intention to improve and standardize the diagnosis of DILI by preventing the introduction of errors and subjective opinions including expert opinion [4]. Indeed, with its specific features, RUCAM is in line with AI principles, promoting RUCAM as an intelligent diagnostic algorithm, ready to be applied by users to provide good quality data for DILI and HILI cases [8]. As it stands, RUCAM includes AI essentials and is based on seven distinct domains comprising key elements that are well defined and provide individual sores [4,18]. Among the RUCAM domains and, for instance, the hepatocellular injury, are the time to onset from the beginning (or the cessation) of the drug/herb use (scores +2 or +1), course of ALT after cessation of the drug/herb (scores +3 to −2), risk factors (scores +1 or 0), concomitant drug(s) and herb(s) (scores 0 to −3), search for alternative causes (scores +2 to −3), knowledge of product hepatotoxicity (scores +2 to 0) and response to unintentional re-exposure (scores +3 to −2) [18]. The scoring range reflects the variability of some criteria and allow for a selection of a precise attribution, avoiding a black or white choice. With +14 down to −9 points, the final score by products indicates the causality level: score ≤0, excluded causality; 1–2, unlikely; 3–5, possible; 6–8, probable; ≥9, highly probable. Needless to say, many details are available in instructions, yet how best to apply the updated RUCAM and how to handle specific questions and conditions may emerge during causality assessment using RUCAM [18,64]. Although described on the LiverTox website, details on how to use RUCAM are not up to date, since the reference to the updated RUCAM [18] and other RUCAM-related publications is also missing [19,64,65]. Therefore, RUCAM benefited from the inclusion of AI principles, making it an ideal algorithm for assessing causality in DILI and HILI cases, which should better be appreciated by the LiverTox database, requiring the substantial actualization of the website text and causality assessment of the presented cases.

The algorithmic structure of RUCAM facilitates the creation of electronic versions or automatized application of this method despite some weaknesses to be corrected [65]. RUCAM is privileged as a structured, transparent, user friendly, objective, and quantitative diagnostic algorithm [18] in line with AI principles [8], specific for hepatic injury caused by drugs and herbs [18] and thereby differs from other CAMs that are not specific to the liver, lack element specification, or do not include a scoring system [18,65]. Indeed, most of them are not suitable for assessing causality in DILI and HILI cases, being subjective because they are based on variable, often divergent opinions of assessors, not validated with a gold standard, not liver specific, and finally not providing causality gradings derived from scored key elements [18,64,65,66]. In other words, most of the other CAMs use a not quantifiable, not transparent, and not valid approach. Therefore, and due to these weaknesses, most other CAMs will likely not be used in the next few years unless the updated RUCAM is fully incorporated, a view commonly shared among DILI and HILI experts in many countries.

Much support for issues of DILI and HILI to be improved and specifically for RUCAM comes annually since several years from US authors [9,10,67,68] working independently from network opinions. They apply the updated RUCAM to reported cases and decided to exclude case reports where they were unable to assess causality with the available information [10], a sensible approach and thus highly appreciated.

As with any well working innovative method in medicine, some background noise is expected for various reasons especially from scientists, who never had established a robust algorithm such as RUCAM. In this context, several unsuccessful attempts by others to add, modify or delete elements, or to upgrade or downgrade scores were frustrating, reducing the user friendly handling of the method, making the method more complex and not validated. The data were, as expected, not published. In fact and as in real life situations, a well running method such as RUCAM should not be changed unless major improvements are expected. If a robust diagnostic biomarker emerges, derived from RUCAM-based DILI and HILI cases, its inclusion in RUCAM is not recommended unless a full and new validation process is done. Thus, it would rather be used in parallel to the updated RUCAM.

Innovative diagnostic biomarkers are needed to support RUCAM-based case evaluations, provided that for the validation process of the new biomarkers, a robust method such as the updated RUCAM is used instead of a weak subjective global introspection method. However, diagnostic biomarkers are currently not available [10,17,65]. Concern has been expressed on studies to DILI that included cases with a “possible” causality grading. Taking into consideration this category of cases reduces the strength of the causal association between the cases and the incriminated drug or herb, or the new biomarker to be validated [65]. Unfortunately, the scientific community involved in DILI and HILI research was confronted with fraud in connection with published data on biomarkers, confirming previous concerns regarding this issue [10] published earlier [17]. It was outlined that the primary fraudulent research has been cited by hundreds of articles as indexed in the PubMed database [10]. As a result, many conclusions of previous original publications and review articles will have to be corrected, and it seems that the official retraction by EMA and the FDA released online on 15 April 2019 has been ignored in some papers published since then and in future publications, currently circulating among reviewers. As a consequence, a follow up of this issue regarding the correct work of authors, reviewers, and editors would be interesting.

## 6. COVID-19, DILI, HILI, and RUCAM

A possible role of drugs and herbs in liver injury found in patients with COVID-19 infection is under discussion and requires innovative approaches [5]. Increased ALT values of variable extent were reported in some patients infected with COVID-19 and require causal attribution [69,70,71,72,73,74,75,76]. The updated RUCAM was used in a single retrospective study of a few patients with COVID-19 infection associated with increased ALT values, but it would be premature to classify some of these cases as actual DILI or HILI due to the low case number and methodological uncertainties [69]. RUCAM-based DILI was also detected in another patient with COVID-19 infection [71]. Required are large prospective studies with clear inclusion criteria. Severe COVID-19 infection is presented as multiorgan failure syndrome affecting mostly older patients with multimorbidity under multipharmacy [69,70,71,72,73,74,75,76,77,78,79], whereas the role of under-reported toxic substance exposure needs further clarification [78]. Organs under consideration are apart from the liver especially the lungs, heart, and brain, raising the question of whether the liver injury can be explained by causes unrelated to drugs or herbs, by a preexisting chronic or coexisting acute liver disease, the virus itself due to its presence in the liver cell, or by other conditions in extrahepatic organs. For instance, the liver injury in patients with COVID-19 infection could be due to systemic sepsis or hypoxia, caused by pulmonary embolism, pneumonia, the injury of the breathing center due to cerebral insult, or cardiac hypoxia through acute cardiac failure or arrhythmias as examples. Therefore, clinical experience can solve these questions, yet certainly not the RUCAM alone.

## 7. Worldwide Use of RUCAM

An earlier editorial suggested to expand our knowledge on DILI by enlarging population analysis with prospective and scoring causality assessment with a CAM like the updated RUCAM to be applied to DILI as well as HILI cases [80]. This was partially achieved considering the recent analysis of a total of 95,865 liver injury cases assessed by RUCAM, published by authors from around the word from 1993 to mid of 2020, and comprising 81,856 DILI cases and 14,029 HILI cases [7]. Therefore, the worldwide use of RUCAM provided additional support by its user-friendly utility.

The primary aim of the study with 95,865 liver injury cases was to evaluate the worldwide use of RUCAM with countries listed in alphabetical order, combined with references and dates of publication provided for the first author of any liver injury case caused by a drug or herb, and listed were the number of injury cases, the drugs or herbs implicated in the liver injury, and comments on the RUCAM-based liver injury cases [7]. Some study cohorts consisted of several hundreds of DILI cases caused by various drugs, and as expected, lacking published causality scores for individual drugs. With respect to scores, the best documented are case reports with a single drug or a few drugs, which can easily be searched for by looking at the number of DILI cases as listed [7]. From a previous study, it is known that most reports on small cohorts published scores of 6 and higher, in line with a probable or highly probable causality grading [19]. Most importantly, if drugs were individually identified, case details can easily be found from the corresponding reference as provided [7]. Liver injury cases with a RUCAM-based score of 5 or less and thereby corresponding to a possible, unlikely, or excluded causality grading are of no value and should not be used for clinical or research purposes.

## 8. Worldwide Annual Growth Trend Analysis of RUCAM Publications

### 8.1. RUCAM-Based DILI Publications

Over the years, starting in 1993, when RUCAM was launched [4] and until 2019, an upward trend of annual RUCAM-based DILI publications was observed with some dips in between [7]. As expected, the publication numbers were low in the years following 1993, likely because studies had to be conceptualized with results available only in the subsequent years. In 2019, 26 publications were counted, with 15 publications in 2020 until the end of June 2020 that were not included in the listing (Figure 1). From 1993 until the middle of 2020, a total of 158 DILI publications were counted that included a total of 81,856 DILI cases [7].

### 8.2. RUCAM-Based HILI Publications

Compared to DILI publications (Figure 1), the overall trend was similar for RUCAM-based HILI publications with an upward tendency from 2004 to 2019, again with some dips in between (Figure 2). Between 1993 and 2003, no HILI publications were recorded, likely due to studies that had to be designed and the fact that HILI was not considered as an important disease at that time. In 2019, 18 publications were counted with four publications in 2020 until the end of June 2020 that were not included in the listing (Figure 2). For the whole year 2020, therefore, at best perhaps eight publications could be anticipated. Overall, 14,029 published RUCAM-based HILI cases were recorded from 1993 until the middle of 2020 [7].

## 9. Scientometric Evaluation and RUCAM

Finally, and most interestingly, are the results of a recent innovative scientometric study conducted by unbiased investigators from China not affiliated to any network. They analyzed the global knowledge base on idiosyncratic DILI in the international DILI scene, provided several rankings, and promoted RUCAM [3]. This study comprehensively analyzed the global knowledge base and specific emerging topics of DILI derived from 1995 publications in 79 countries and regions. They found an annual growth trend of reports between 2010 and 2019 and anticipated almost 340 studies to be published in 2020 [3]. These findings follow the rise of publications on DILI and HILI cases with causality assessment using RUCAM (Figure 1 and Figure 2) [7]. This study confirmed the high worldwide interest in DILI publications and identified a total of 1995 DILI studies published between 2010 and 2019, although the information on the causality assessment method was not provided [3]. This analysis also showed the top 10 countries involved in DILI research: US, China, Japan, Germany, UK, Spain, France, the Netherlands, Sweden, and Canada. In addition, many interesting aspects on DILI were comprehensively discussed with focus on definition, incidence rate, clinical characteristics, etiology or pathogenesis such as the role of the innate immune system, the regulation of cell-death pathways, susceptible human leukocyte antigen (HLA) identification, or causality assessment criteria and methods, all topics that are considered as the knowledge base for DILI research [3].

Regarding RUCAM, not unexpectedly and highly appreciated, the original RUCAM of 1993 by G. Danan and C. Benichou [4] was highlighted as a publication that was frequently co-cited (*n* = 256) and ranked first in the category of the top 10 co-cited references related to DILI research according to the scientometric investigation [3]. This study also mentioned a group of the University of Michigan (R.J. Fontana) and Frankfurt/Main (R. Teschke), who may have significant influence on DILI research with more publications (*n* = 46; *n* = 39) and co-citations (*n* = 382; *n* = 945) according to the Chinese experts [3]. Other experts included in the Chinese report [3] are to be mentioned among the top 10 authors involved in DILI-related studies: P.B. Watkins, R.J. Fontana, R. Teschke, T. Yokoi, R.J. Andrade, N. Chalasani, B.K. Park, D.E. Kleiner, W.M. Lee, and J. Uetrecht. The node sizes of P.B. Watkins and R.J. Fontana are described as larger since most of their publications were derived from a US network (DILIN), whereas R. Teschke is correctly described as not being part of any network [3]. For the latter researcher, independency from a network and governmental financial support is important, allowing for critical analyses and comments [5,6,7,8]. Overall, the scientometric evaluation is worth being read by experts in the field, also because it encourages further studies on liver injury induced by drugs.

## 10. Conclusions

RUCAM is a quantitative, structured, transparent, user-friendly, validated, and liver injury-specific diagnostic algorithm, based on principles of artificial intelligence to resolve complex issues as diseases like DILI and HILI are. For diagnostic purposes, RUCAM was used worldwide, between 1993 and mid-2020, to assess causality in 81,856 DILI and 14,029 HILI cases. Based on defined elements and their scores, RUCAM provides a final causality grading for each case. The elements of RUCAM were agreed upon by international experts following consensus meetings, and scores were adequately validated making RUCAM a robust algorithm. There were several unsuccessful attempts by others to add or delete elements, to modify criteria, or to upgrade or downgrade scores, but results were frustrating, reduced user-friendly application, and made the method more complex. Each of the new modified methods proved not being validated, and results were not published, as expected by experts familiar with RUCAM. After the dilemma with EMA and the US FDA, new and innovative robust diagnostic biomarkers are welcomed, and their development should be supported, now hopefully derived from DILI and HILI cases based on the updated RUCAM. However, their inclusion in RUCAM is not recommended unless they are properly validated and their weight adequately assigned, and they should be used better in parallel to the updated RUCAM for a better assessment in the benefit of the patients. In the future, RUCAM-based DILI and HILI cases are valuable tools establishing valid genetic or non-genetic risk factors like drug dose, drug metabolism, or lipophilicity, and hopefully mechanistic steps leading to the liver injury.

## Figures and Tables

**Figure 1 diagnostics-11-00458-f001:**
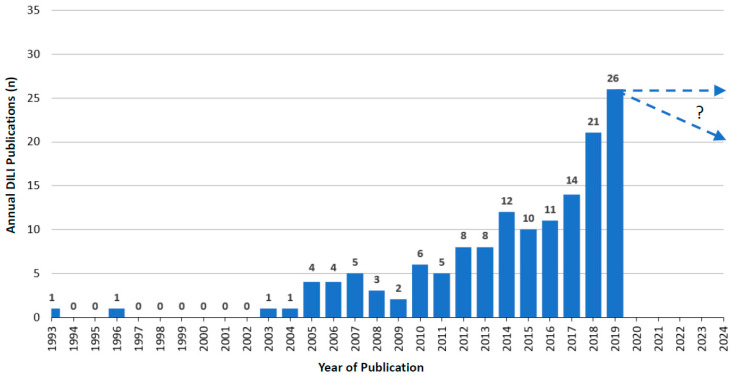
Annual publications of DILI cases assessed for causality by RUCAM as reported since 1993. This figure is derived from a previous report of the authors [7], which provides many details.

**Figure 2 diagnostics-11-00458-f002:**
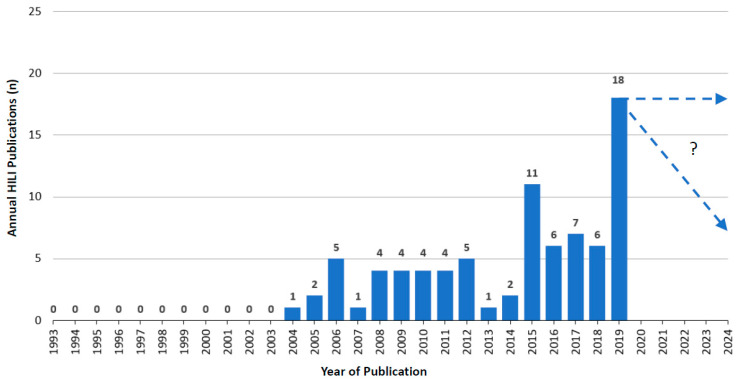
Annual publications of HILI cases assessed for causality by RUCAM as reported since 1993. This figure is derived from a previous report of the authors [7], which provides many details.

**Table 1 diagnostics-11-00458-t001:** Criteria of liver adaptation and liver injury classification.

Mechanistic Background	Thresholds of Liver Tests	Criteria and Characteristic Features	Recommended Description
Adaptive	ALT ≤ 5 × ULN ALP ≤ 2 × ULN	Develops at the recommended daily dose Presumably the majority of drugs have the potency of causing rare but clinically not apparent liver adaptation Normalization or stabilization of liver tests is commonly observed whether the drug is discontinued or continued	Liver adaptation
Idiosyncratic	ALT ≥ 5 × ULN ALP ≥ 2 × ULN	Develops at recommended daily doses Cessation of drug use is mandatory and immediate Worsening if drug is continued Most drugs cause idiosyncratic DILI, often called DILI in short if not specified Risk of acute liver failure	Idiosyncratic DILI
Intrinsic	ALT ≥ 5 × ULN ALP ≥ 2 × ULN	Due to drug overdose Only few drugs are known for causing intrinsic DILI, antidotes may be available Risk of acute liver failure	Intrinsic DILI

Criteria are listed for DILI but are also applicable to herb-induced liver injury (HILI), modified from a previous report [19]. Abbreviations: ALP, alkaline phosphatase; ALT, alanine aminotransferase; DILI, drug-induced liver injury; ULN, upper limit of normal.

**Table 2 diagnostics-11-00458-t002:** Selected reports with diagnostic algorithms based on principles of artificial intelligence.

Reporting Country	Year of Publication	Diseases and Applications	First Author
France	1993	DILI and RUCAM	Danan [4]
France	1993	DILI and RUCAM	Bénichou [26]
UK	2006	Haematuria	Rodgers [37]
Germany	2008	Autoimmune hepatitis	Hennes [38]
US	2009	Heart-lung transplantation	Oztekin [39]
US	2011	Gaucher disease	Mistry [40]
US	2012	Ankle injuries	Okanobo [41]
Austria	2012	Tuberculosis	Ratzinger [42]
Germany	2013	Hepatocellular carcinoma	Schirmacher [43]
Italy	2014	Gaucher disease	Di Rocco [44]
Spain	2014	DILI, RUCAM, and acute liver failure	Robles-Diaz [45]
Italy	2016	Acute coronary syndrome	Cervellin [46]
Italy	2016	Autoimmune encephalitis	Damato [47]
Australia	2016	Giant cell arteritis	George [48]
Germany	2016	Charcot–Marie–Tooth neuropathies	Rudnik-Schöneborn [49]
US	2017	Cardiopulmonary diseases	Ghamloush [50]
US	2017	Cardiopulmonary diseases	Ley [51]
Hungary	2017	Osseous metastatic cancer diseases	Szendrői [52]
China	2018	Pediatric otitis media	Tran [53]
US	2018	B-cell lymphomas	Wang [54]
Netherlands	2019	Cerebellar ataxia	Brandsma [55]
Korea	2019	Pathology diagnostics	Chang [56]
Germany	2019	Central ocular motor disorders	Kraus [57]
Canada	2019	Heart transplantation	Parkes [58]
Germany	2019	Heart failure	Pieske [59]
US	2019	Cardiovascular diseases	Singh [60]
Austria	2019	Mast cell activation syndrome	Valent [61]
India	2020	Various	Kamdar [62]
UK	2020	Febrile illnesses	Pokharel [63]

Abbreviations: DILI, drug-induced liver injury; RUCAM, Roussel Uclaf Causality Assessment Method.

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
