# Peer review of "Idiosyncratic Drug-Induced Liver Injury (DILI) and Herb-Induced Liver Injury (HILI): Diagnostic Algorithm Based on the Quantitative Roussel Uclaf Causality Assessment Method (RUCAM)"

_diagnostics, 2021, doi:10.3390/diagnostics11030458_

Round 1

Reviewer 1 Report

The current version of the manuscript has improved. However, there is still a scope to further hone the manuscript as author has not considered all the suggestions. It is advised that the previous comments be taken into account so that this manuscript can be an informative reading to advance thinking.

Author Response

Thank you for your patience. Your points were now considered on: Lines 12-13, first sentence of introduction, and lines 42-46, 49-57, 203-218, 267, 278, 297,302-303, 322-336, 361, 408-409.

Reviewer 2 Report

In the manuscript, the authors present a review regarding the clinical importance of a diagnostic algorithm in cases of drug induced liver injury and herbs induced liver injury. In my opinion, it is an interesting manuscript that can be published. Also, in order to improve the quality of the manuscript some changes have to be done. My observations are :

  • Is not correct to use abbreviations in the title of the manuscript.
  • There are some grammar and spelling errors in English. For example, in the line 58, the correct term is COVID-19 instead of CORVID-19. The manuscript must be polished by an native English speaker.
  • please introduce in the main manuscript a detailed analysis of the studies that were introduced in the study in which the RUCAN algorithm was used.. (the scoring range)

Author Response

Thank you for your suggestions:

Abbreviations in the title were expanded as suggested. Line 58 correction was done. A native English speaker reexamined the text, found only a few examples, which were now corrected in the text. Detailled analysis was included on L 332-336.

Round 2

Reviewer 2 Report

In my opinion, it is an interesting manuscript. The manuscript has been reviewed before and the authors changed the manuscript according to the previous reviewers indications. That is why, I think that this manuscript can be published in this form.

This manuscript is a resubmission of an earlier submission. The following is a list of the peer review reports and author responses from that submission.

Round 1

Reviewer 1 Report

Recommendation

This paper provides a comprehensive description of the development of RUCAM, and clarifies the problems encountered in the development of RUCAM and related work, which can provide an effective way for other researchers to understand more comprehensively the work related to RUCAM. However, the article still has some problems as follows.

1、The article lacks a certain amount of commentary, making it lack depth.

2、The essay is more of a declarative overview, with insufficient summary induction.

3、The citations in the article literature give the impression of stacked citations.

4、The article mainly analyzes idiosyncratic DILI and HILI cases with respect to causality assessment based on the quantitative RUCAM, but the papers cited in the article are more simple statements, lacking some summary and induction, which makes the research value and innovation of the article reduced.

5、Chapter 6, the combination with AI should be an important development direction of RUCAM, however, the article lacks some new insights from the authors, such as "what new opportunities and advantages this combination can bring for the development of RUCAM", which is also an important problem of the whole article.

Reviewer 2 Report

The manuscript is well-writen and easy to follow. However, the content has been mostly covered by the other publication [1]. For instance, Figure 1 and 2 have been reported in [1] as Figure 2 and 4, respectively. Therefore, it might be better not accepted.

[1] Teschke, R. and Danan, G., 2020. Worldwide Use of RUCAM for Causality Assessment in 81,856 Idiosyncratic DILI and 14,029 HILI Cases Published 1993–Mid 2020: A Comprehensive Analysis. Medicines, 7(10), p.62.

Reviewer 3 Report

diagnostics-1073694

Title:  Idiosyncratic DILI and HILI: Diagnostic algorithm based on the quantitative Roussel Uclaf Causality Assessment Method (RUCAM)

Rolf Teschke *, Gaby Danan

Overview and general recommendation:

The paper describes the extensive use of diagnostic algorithm RUCAM (Roussel Uclaf Causality Assessment Method) to assess causality in suspected cases of DILI (drug induced liver injury) and HILI (herb induced liver injury). It claims that the algorithm is in line with AI principles.

Following are some of the major concerns regarding the paper:

Comments:

  • The aim of the paper is not clearly mentioned in the abstract. Is the paper proving RUCAM is an AI technique?
  • It is understood that RUCAM is a well-established method known for its quantitative features with specific data elements that are individually scored to obtain final causality grading. Too much emphasis is made to link RUCAM to AI.
  • Section 2 : Literature search and source, the terms AI or Artificial Intelligence is not used, which makes this paper not distinct from the author’s previous similar publications (e.g. Reference 7).
  • Much of information in section 3, 7, 8 and Tables 1 and 2 are already mentioned in Reference 7. So it is not needed to be explained in this paper again.
  • Table 2 title says selected reports with algorithms based on principles of AI. But the tabular column gives unwanted information like year of publication, country and author name instead of providing information about what AI principle is the author talking about in the papers.
  • The conclusion again recommends RUCAM, not mentioning any word about AI.

Overall, I recommend the paper to be rewritten, excluding repeated information from previous publications and adding more information on how RUCAM is based on principles of Artificial Intelligence.

Reviewer 4 Report

In this review paper, Idiosyncratic drug induced liver injury (DILI) and herb induced liver injury (HILI) cases were examined for causality evaluation based on quantitative RUCAM as the commonly used diagnostic algorithm, early developed in accordance with the principles of AI. The review features an important research area on causality evaluation in DILI and HILI using quantitative RUCAM. Though the review highlighted the specific examples but lack mechanistic insights. There is sufficient scope to revise this review further to feature a balanced review report with important highlights to advance the thinking in the field.

The introduction should be expended with what extra information this manuscript provides to advance understanding in the relevant discipline.

Although there is enough scope to include figures and relevant infographics to clarify the significant summaries discussed, there is no paper included. It is recommended that the related infographics be used as figures to provide a visual explanation of the essential summaries and frameworks for this analysis and interesting reading.

Manuscript should be carefully proofread to fix moderate language errors and typos.